# Facile Mechanochemical Synthesis of Nickel/Graphene Oxide Nanocomposites with Unique and Tunable Morphology: Applications in Heterogeneous Catalysis and Supercapacitors

**Mayakrishnan Gopiraman [1]**, **Somasundaram Saravanamoorthy [2]**, **Dian Deng [3]**,
**Andivelu Ilangovan [2]**, **Ick Soo Kim [3]** and **Ill Min Chung [1],\***

[1] Department of Applied Bioscience, College of Life & Environment Science, Konkuk University,
120 Neungdong-ro, Gwangjin-gu, Seoul 05029, Korea; gopiramannitt@gmail.com

[2] School of Chemistry, Bharathidasan University, Tiruchirappalli, Tamil Nadu 620 024, India;
saraartudc@gmail.com (S.S.); ilangovan@bdu.ac.in (A.I.)

[3] Nano Fusion Technology Research Group, Division of Frontier Fibers, Institute for Fiber Engineering (IFES),
Interdisciplinary Cluster for Cutting Edge Research (ICCER), Shinshu University, Tokida 3-15-1, Ueda,
Nagano Prefecture 386-8567, Japan; dian.deng.su@gmail.com (D.D.); kimicksoo.gr@gmail.com (I.S.K.)

\* Correspondence: imcim@konkuk.ac.kr or illminchung@gmail.com; Tel.: +82-02-450-3730;
Fax: +82-02-446-7856

**Abstract:** In this study, a very simple and highly effective mechanochemical preparation method was developed for the preparation of Ni nanoparticles supported graphene oxide (GO) nanocomposites (Ni/GO, where Ni is a composition of $Ni(OH)_2$, $NiOOH$, $NiO$, $Ni_2O_3$ and $NiO_2$), 3 wt% NiO/GO (Ni/GO-1) and 8 wt% NiO/GO(Ni/GO-2). The developed method is not only very simple and efficient, but also, the morphology of Ni/GO nanocomposites can be tuned by simply varying the metal loading. Morphology and specific surface area of the resultant Ni/GO nanocomposites were investigated by mean of AFM, HR-TEM and BET. Chemical sate and factual content of Ni in Ni/GO-1 and Ni/GO-2, and the presence of defective sites in Ni-nanocomposites were investigated in detail. To our delight, the prepared Ni/GO-2 demonstrated superior catalytic activity toward the reduction of 2- and 4-nitrophenol in water with high rate constant (kapp) of $35.4 \times 10^{-3}$ $s^{-1}$. To the best of our knowledge, this is the best efficient Ni-based graphene nanocomposites for the reduction of 2- and 4-NP reported to date. The Ni/GO-1 and Ni/GO-2 demonstrated an excellent reusability; no loss in its catalytic activity was noticed, even after 10th cycle. Surprisingly the Ni/GO-2 as electrode material exhibited an excellent specific capacitance of 461 F/g in 6 M KOH at a scan rate of 5 mV. Moreover, the Ni/GO nanocomposites were found to possess poor electrical resistance and high stability (no significant change in the specific capacitance even after 1000 cycles).

**Keywords:** graphene oxide; mechanochemical synthesis; Ni-nanocomposites; reduction; reusable; supercapacitor

---

## 1. Introduction

Graphene, a new class of two-dimensional (2D) carbon materials (CMs) with a honeycomb structure, has become a 'rising star' in the field of material science due to its excellent physicochemical properties [1,2]. A unique 2D structure, huge surface area, excellent conductivity and high mechanical strength offers graphene as a suitable support material for the development of multifunctional graphene-based nanocomposites [3]. A considerable number of graphene-based nanocomposites (mainly based on transition metals) has been developed for nanoelectronics [4], sensors [5],

supercapacitors [6,7], batteries [8], fuel cells [9], photovoltaics [10], catalysis [11,12], and biomedical applications [13]. Recently, very simple and low-cost nickel/graphenenanocomposites have received much attention due to their use in numerous practical applications [14,15]. Salimian et al., [16] prepared reduced graphene oxide/spiky nickel nanocomposite and it was used for nanoelectronic applications. Zhao and co-workers [17] utilized magnetic Ni@graphene nanocomposites for efficient removal organic dye under ultrasound. Very recently, Zhou et al., [18] obtained Ni/graphene nanocomposite for hydrogen storage applications. As result, several preparation methods have been developed to achieve Ni/graphene nanocomposites with unique physicochemical properties in a reproducible and controlled manner [19,20]. Most preparation methods involve either expensive or toxic reagents such as $NaBH_4$, $N_2H_4$, NaOH, KOH, octadecylamine, urea and ethylene glycol. For instance, Chen and co-workers [21] synthesized nickel/graphene hybrids under microwave irradiation by using $NaHB_4$ as a reducing agent and NaOH as an alkaline medium. Kollu et al. [22] adopted one-pot solvothermal method for preparation of RGO-Ni/NF nanocomposite in which ethylene glycol was used to avoid unnecessary aggregation or growth of Ni NPs. Similarly, Ji et al. [23] prepared RGO/Ni nanocomposites by using $NaHB_4$ and NaOH as a reducing agent and alkaline medium. Complexation of Ni with urea followed by reduction with $N_2H_4$ made it possible to reduce GO/Ni nanocomposites [24]. Electrochemical deposition method is also widely employed for Ni/graphene preparation. Ren et al. [25] deposited Ni NPs on graphene surface by electrochemical deposition method. Other methods which require no additional reagents and organic solvents were also developed for the preparation of Ni/graphene nanocomposites. For example, conventional thermal chemical vapor deposition (CVD) system was used to prepare Ni/graphene nanocomposites with excellent electromagnetic and electrocatalytic properties by Cao and co-workers [26]. However, the aforementioned techniques are either expensive or require toxic reagents.

Until now, very few environmentally friendly preparation methods have been reported for the preparation of Ni/graphene nanocomposites. Generally, the preparation method is very important, since it determinesthe key properties of nanocomposites. According to Yang et al. [27], defective graphene often exhibits enhanced reactivity due to existence of unsaturated coordination number of carbon in these locations or to functional group attached to the carbon. Mechanical ball milling or etching are typical methods which can create more defect sites in graphene [28]. Zaid et al. [29] hybrid layered graphene/nickel nanocomposite by ball-milling graphene and Ni NPs at a rotatory speed of 400 rpm for 20 h. In spite of enhanced electrocatalytic performance, dispersion and attachment of Ni NPs on graphene surface were found to be poor. In addition, the shape and size of Ni NPs were observedto be large and uneven, respectively. So far, very few studies have focused for the development of multifunctional carbon nanocomposites by this simple 'mix and heat' or 'dry synthesis' method [30–32]. We found that the resultant nanocomposites are highly efficient in catalytic reactions. However, achieving better morphology of nanocomposites, particularly at high metal loading, is always a challenging task by the 'dry synthesis' method. Herein, we report a simple and highly efficient mechanochemical synthesis of graphene oxide/nickel nanocomposites (Ni/GO-1 and Ni/GO-2, where Ni is a composition of $Ni(OH)_2$, NiOOH, NiO, $Ni_2O_3$ and $NiO_2$). The Ni/GO-1 and Ni/GO-2 were studied in detail by means of TEM, SEM-EDS, XRD, Raman, XPS and BET. Catalytic activity of the Ni/GO-1 and Ni/GO-2 was studied toward the reduction of 4-nitrophenol (4-NP) and 2-nitrophenol (2-NP). Supercapitor behavior of Ni/GO-1 and Ni/GO-2 was also studied in detail.

## 2. Experimental Section

### 2.1. Materials

Industrial-quality graphene oxide (GO) with a purity of about >99 wt%, surface area of >600 $m^2$/g and thickness of ≤3.0 nm was purchased from ACS Materials (Pasadena, CA, USA) and used as received. $Ni(acac)_2$ (97%), sodium borohydride ($NaBH_4$), 4-nitrophenol (4-NP) and 2-nitrophenol (2-NP) were

purchased from Sigma Aldrich. All other chemicals (nafion solution, isopropanol and KOH) were purchased from Wako Pure Chemicals (Japan) or Sigma-Aldrich (USA) and used without purification.

### 2.2. Preparation of Ni/GO Nanocomposites

A simple mechanochemical synthesis was developed for the preparation of Ni/GO-1 and Ni/GO-2. Neither reducing agent nor capping agent was used for the preparation. In a typical preparation, 30 mg of $Ni(acac)_2$ was dissolved in 10 mL of DMF. On other hand, 200 mg of GO was taken in a mortar. The prepared $Ni(acac)_2$ solution was mixed with GO taken in the mortar and $GO/Ni(acac)_2$ paste was obtained by pestle-grinding for 30 min. Subsequently, DMF present in the resultant paste was slowly evaporated by using oven at 130 °C for several hours. Finally, the dried $GO/Ni(acac)_2$ was further grinded for 15 min using mortar-pestle and the resultant mixture was calcinated under $N_2$ atmosphere at 400 °C for 3 h. Similarly, Ni/GO-2 was prepared by using 90 mg of $Ni(acac)_2$ and 200 mg of GO. In order to investigate the capability of the present preparation method, high metal loading was also done. Ni/GO nanocomposite with 15 wt% of Ni was also prepared (results are provided in the supporting information). Figure 1 shows schematic illustration for the preparation of Ni/GO-1 and Ni/GO-2.

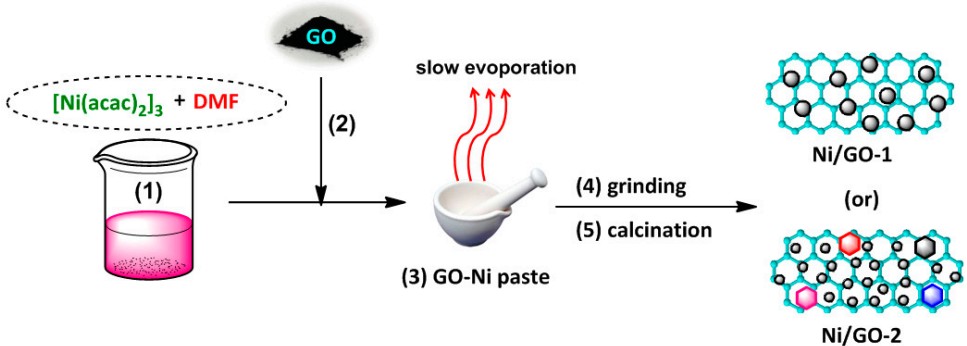

**Figure 1.** Schematic illustration for the preparation of Ni/GO-1 and Ni/GO-2.

### 2.3. Characterization

Transmission electron microscope (TEM, JEOL JEM-2100F) with accelerating voltage of 200 kV was used to study the morphology of Ni/GO-1 and Ni/GO-2. The content of Ni in Ni/GO-1 and Ni/GO-2 was determined using scanning electron microscope-energy dispersive spectroscope (SEM-EDS, Hitachi 3000H SEM) and inductively coupled plasma-mass spectrometer (ICP-MS, 7500CS, Agilent). Raman spectroscopy (Hololab 5000, Kaiser Optical Systems Inc., USA) was used to investigate the metal-support interaction in Ni/GO-1 and Ni/GO-2. Powder X-ray diffraction (XRD) experiment was performed using a Rotaflex RTP300 (Rigaku Co., Japan) diffractometer at 50 kV and 200 mA. X-ray photoelectron spectroscopy (XPS) was recorded on Kratos Axis-Ultra DLD, Kratos Analytical Ltd, Japan. Brunauer-Emmett-Teller (BET) method (BELSORP-max; BEL Japan, Inc.) was adopted to determine the specific surface area of the Ni-GO composites. Catalytic performance of the Ni-GO composites was investigated by using Ultraviolet-visible (UV-vis, Shimadzu UV-2600 spectrophotometer). Electrochemical measurements (VersaSTAT-4 potentiostat (Ametek, USA)) were performed by using Ni-GO composites as working electrode at a scan rate ranging from 5 to 100 mV s$^{-1}$.

### 2.4. Procedure for the Reduction of Nitrophenols

The catalytic activity of Ni/GO nanocomposites was studied in the reduction of 2- and 4-nitrophenol. A mixture of 2-nitrophenol (2-NP) or 4-nitrophenol (4-NP) (80 µL, 0.01 M) and $NaBH_4$ (4 mL, 0.015 M) was magnetically stirred at 27 °C. Followed by, Ni/GO nanocomposites (0.25, 0.50, 0.75, 1, 1.5 or 2 mg) was added to the above mixture and stirred under open air atmosphere for 30 s. The reaction mixture was sampled and measured at regular time intervals. The absorption spectra were recorded for the

reaction mixture in the range of 250–500 nm. After use, the Ni/GO nanocomposites was separated out from the reaction mixture and reused several times. For comparison, the UV-vis spectra were taken for the aqueous 2-NP and 4-NP solution with and without addition of $NaBH_4$.

*2.5. Electrochemical Studies*

The obtained Ni/GO nanocomposites were used as electrode materials for supercapacitor application. In a typical electrochemical measurement, aqueous solution of 6 M KOH was used as electrolyte. The sweep potential range was adjusted from −1.0 to 1.0 V [vs. Ag/AgCl] in an electrochemical cell with three-electrode system: Pt wire as counter electrode, Ag/AgCl as reference electrode and the Ni/GO nanocomposites (Ni/GO-1 and Ni/GO-2) as working electrode. VersaStat4 potentiostat device was employed to control the above mentioned three electrode system. In a typical preparation of a working electrode, a mixture of 1 mg of Ni/GO nanocomposites, 20 μLNafion solution (5 wt%) and 400 μL isopropanol was sonicated for 2 h at room temperature. Then a 45 μL of the above mixture was taken and carefully deposited on the active area of the glassy carbon electrode. Finally, the glassy carbon electrode was kept in an oven at 80 °C for 30 min to remove the solvent. The galvanostatic charge–discharge measurement was performed at 0.5–10 A $g^{-1}$ over a voltage range of −0.4–0.6 V vs. Ag/AgCl. The specific capacitance of the Ni-nanocomposites was calculated by using the following equation:

$$C = I\Delta t/ m\Delta V \tag{1}$$

where I (A) and Δt (s) refer the discharge current and discharge time respectively and m (g) is the mass of Ni-nanocomposites in the working electrode, and ΔV (V) is the potential window during the discharge process.

## 3. Results and Discussion

*3.1. Characterization of Ni/GO Nanocomposites*

The prepared Ni/GO nanocomposites were characterized in detail by means of various spectral and microscopic techniques. A good nanocatalyst should have properties such as strong metal-support interaction, good conductivity, high surface area, good surface morphology and smaller size of nanoparticles with narrow particles size distribution [33]. In general, the carbon material including graphene are highly hydrophobic; therefore, prior to the decoration of metal nanoparticles, the surface of graphene should be modified with functional groups containing O, N or S atoms [34]. These functional groups might act as additional anchoring sites for the nanoparticles and help to achieve the better dispersion and adhesion of nanoparticles on graphene surface. In addition, most of the developed methods for the preparation of graphene-metal nanocomposites require expensive capping agents to control the particles size, whereas, the present method does not require any pretreatment for graphene oxide and still it satisfies the important requirements of the efficient catalyst. In addition, morphology of carbon-supported metal catalysts, particularly, at high metal loading is obviously hard to control, whereas, the present method achieved it with any reducing agent and capping agent.

Figure 2 shows the TEM images and NiO nanoparticles size distribution of Ni/GO-1 and Ni/GO-2. Interestingly, regardless of metal loadings, the NiO particles size was found to be highly uniform and the particles dispersed homogeneously on the GO surface. At low Ni loading of ~3 wt%, the Ni-particle size was about ~20.5 nm with narrow particles size distribution. At high metal loading of ~8 wt%, a unique see-island like morphology was noticed in which two different sizes of Ni-particles with uniform size distribution were clearly seen; (1) smaller nanoparticles with ~2.9 nm and (2) bigger nanoparticles with ~25.5 nm. The smaller NiO nanoparticles were spherical in shape, whereas, the shape of bigger nanoparticles was found to be irregular. A unique hexagonal shape of some big NiO nanoparticles was confirmed by magnified TEM images (Figure 2f). Moreover, the TEM images confirmed that the GO sheets are not aggregated and highly maintained the single layer nature of the GO. The Ni/GO nanocomposite with 15 wt% was also prepared to realize the adoptability of the

present preparation method (TEM images are given in Supporting Information, Figure S1). To our delight, the Ni/GO nanocomposite with 15 wt% was also found to have good morphology. TEM images confirmed the homogenous dispersion of NiO nanoparticles with average particle size of around 19 nm. One dimensional (1D) AFM images and their corresponding three-dimensional (3D) projections were taken for Ni/GO-1, Ni/GO-2 and Ni/GO-3 (Figure 3 and Figure S2). The results clearly shows that the NiO nanoparticles supported on the surface of GO. In case of Ni/GO-1, NiO nanoparticles size of ~20 nm with narrow particles size distribution was observed. Whereas AFM images of Ni/GO-2 confirmed the decoration of two different sizes of NiO nanoparticles; smaller nanoparticles with ~3 nm and the bigger nanoparticles with ~25 nm. The results are in good agreement with the TEM images.

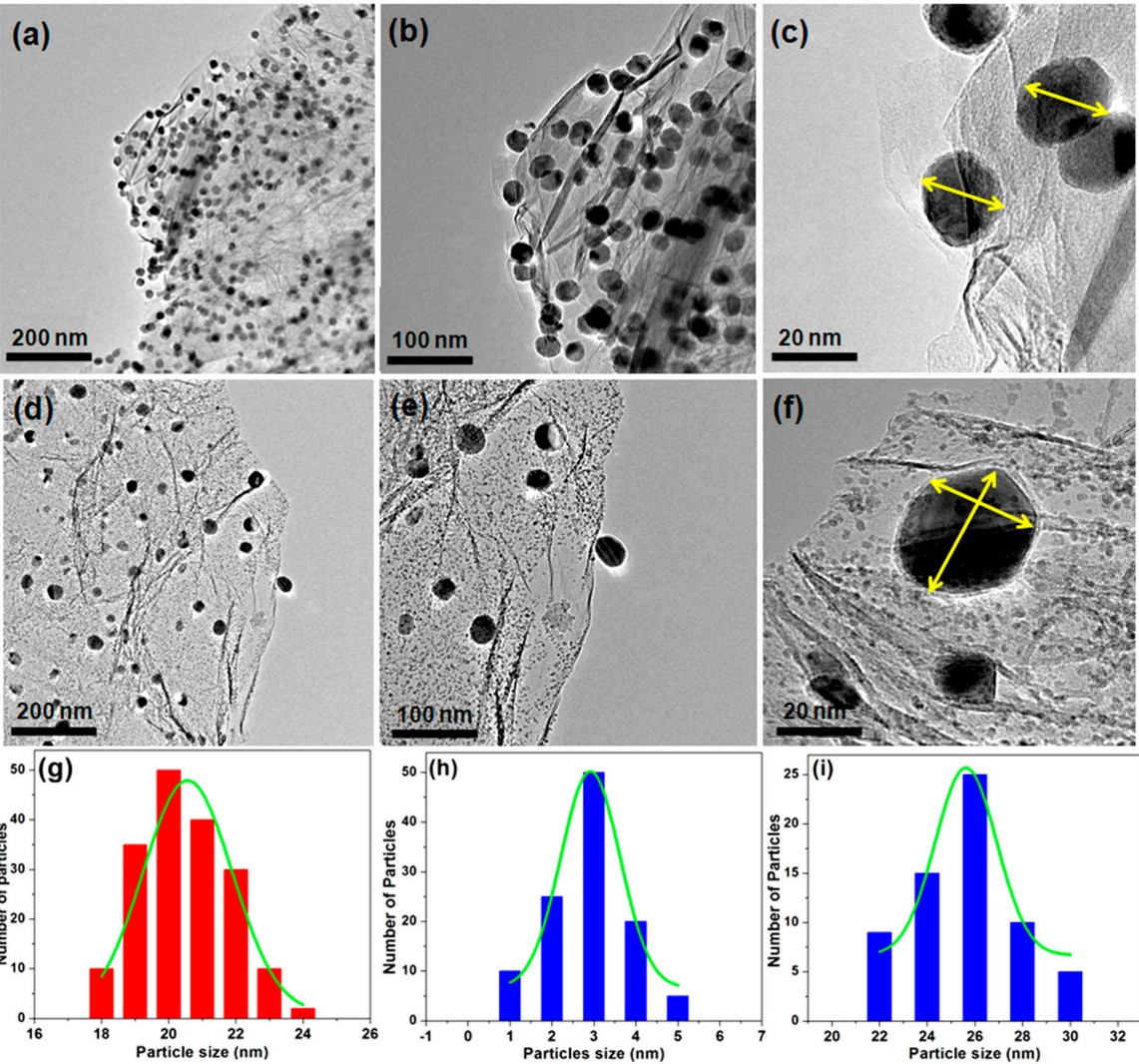

**Figure 2.** TEM images of (**a**–**c**) Ni/GO-1 and (**d**–**f**) Ni/GO-2, and particle-size distribution histogram of NiO nanoparticles in (**g**) Ni/GO-1 and (**h,i**) Ni/GO-2.

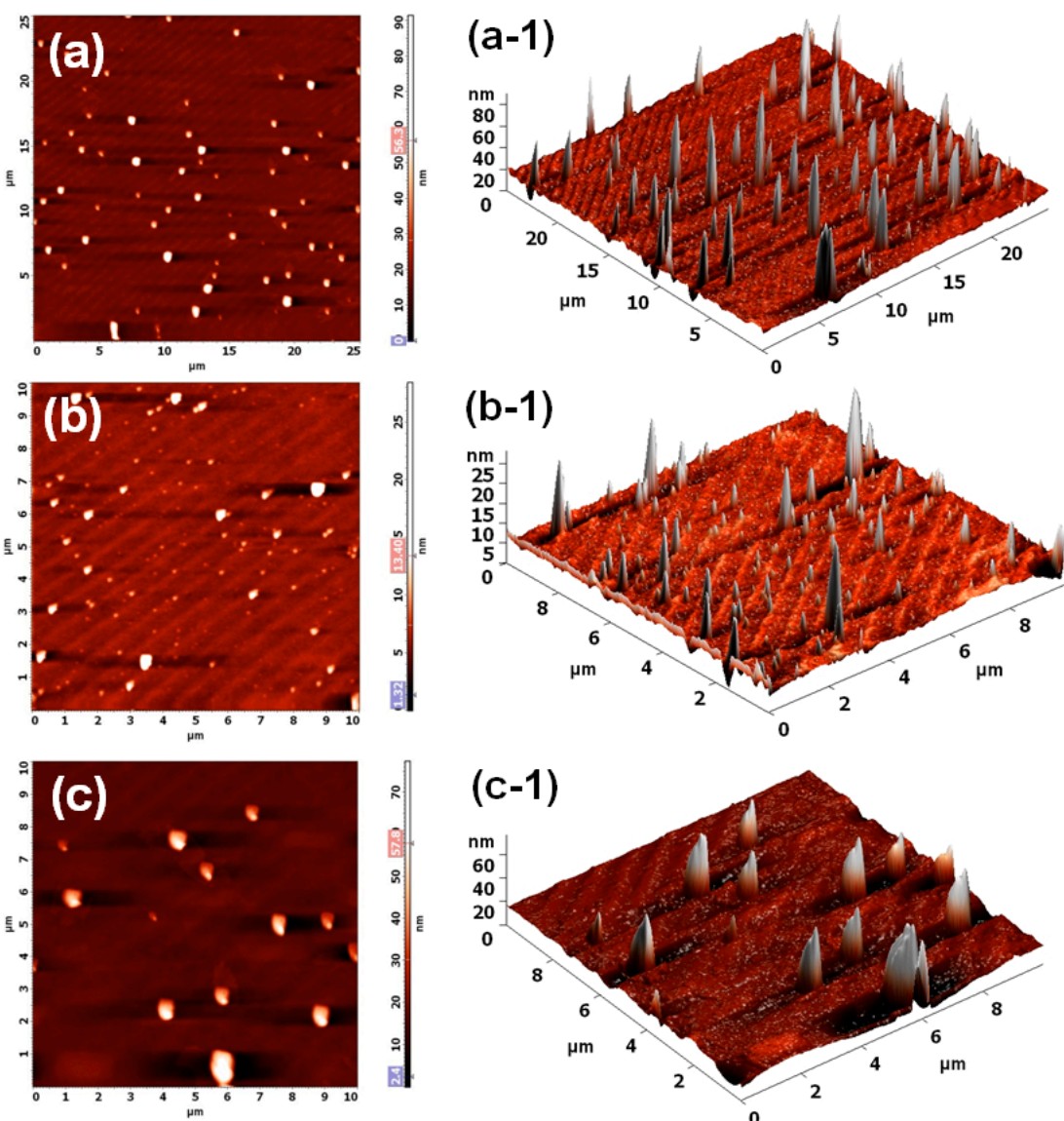

**Figure 3.** AFM images of (**a**,(**a-1**)) Ni/GO-1, (**b**,(**b-1**)) Ni/GO-2, and (**c**,(**c-1**)) Ni/GO with 15 wt% of Ni (Ni/GO-3); performed directly on the surface of samples and their three-dimensional projections.

Five different places were chosen to record SEM-EDS data and the average content of C, O and Ni in Ni/GO-1 and Ni/GO-2 was determined. Figure 4 presents the representative SEM and corresponding EDS spectra of Ni/GO-1 and Ni/GO-2. Weight percentage (wt%) of C, O and Ni in Ni/GO-1 was found to be 78, 19, and 3 respectively. Similarly, in Ni/GO-2, the content of Ni, C and O was determined to be 8, 70 and 22 wt% respectively. Interestingly, no other elements or impurities except C, O and Ni were detected in the catalysts, which show the excellence and reliability of the present method. ICP-MS was taken in order to verify the Ni-loading in Ni/GO nanocomposites. The factual loading of Ni is 2.8 and 8.3 for Ni/GO-1 and Ni/GO-2 respectively. The EDS result agrees well with the ICP-MS values.

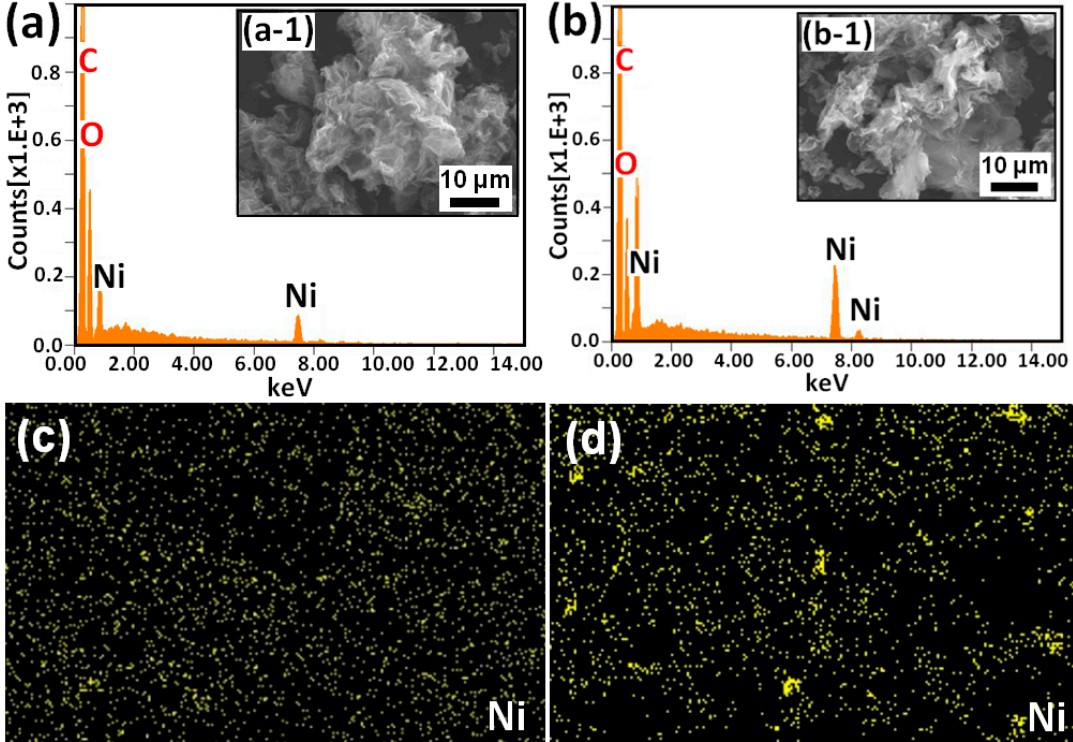

**Figure 4.** SEM images and EDS spectra of (**a**,(**a-1**)) Ni/GO-1 and (**b**,(**b-1**)) Ni/GO-2; and the corresponding Ni-mapping of (**c**) Ni/GO-1 and (**d**) Ni/GO-2.

Raman spectra were recorded for GO, Ni/GO-1 and Ni/GO-2, under 514.5 nm excitation over the Raman shift interval of 500–2500 cm$^{-1}$; the results are presented in Figure 5a. Two main Raman features such as well-defined D-band line at ~1345 cm$^{-1}$ and G-band line at ~1570 cm$^{-1}$ were clearly observed. The D-band line is related to the amount of defect sites in graphene, whereas the G-band line represents the relative degree of graphitization [35]. $I_D/I_G$ ratio was calculated for GO, Ni/GO-1 and Ni/GO-2 from the intensity of D band and G band. The $I_D/I_G$ ratio was calculated to be 0.87, 0.89 and 0.90 for GO, Ni/GO-1 and Ni/GO-2 respectively.It was observed that the $I_D/I_G$ ratio of GO slightly increased after metal decoration (from 0.87 to 0.90) which indicates a possible attachment of Ni nanoparticles on the GO surface. To further, XRD spectra were recorded for GO, Ni/GO-1 and Ni/GO-2 (Figure 5b). A broad peak was observed at 2θ = ~25° for all the three samples, which attributed to the (002) plane of hexagonal graphite structure. It is known that the metallic Ni nanoparticles show strong diffraction peaks at 2$\theta$ = 44.45°, Ni(111); 51.73°, Ni(200); and 76.84°, Ni(220). In the present study, the new peaks at 2θ =~45° and 2θ =~52° correspond to Ni-oxides such as NiO, $Ni_2O_3$ and $NiO_2$ were clearly seen [36]. The intensity and broadness of these peaks reveal the very small size and nanocrystalline nature of the Ni-oxide particles [37].

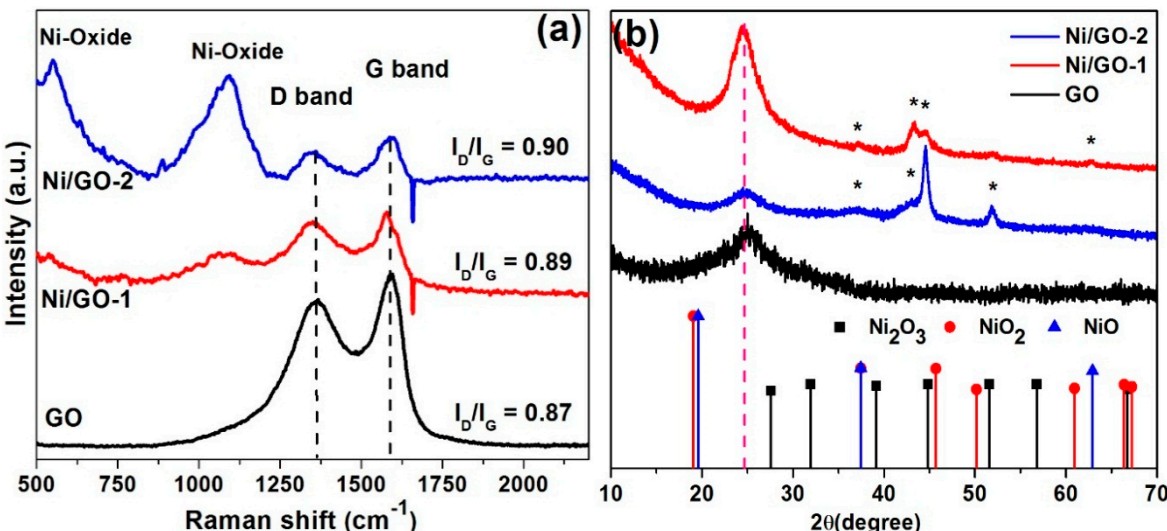

**Figure 5.** (**a**) Raman spectra and (**b**) XRD pattern of GO, Ni/GO-1 and Ni/GO-2.

Figures 6 and 7 show XPS spectra of GO, Ni/GO-1 and Ni/GO-2. Chemical state of Ni in Ni/GO nanocomposites was studied in detail. Figure 6a shows two intense peaks at 284.3 and 533.4 eV, corresponding to C 1s and O 1s, respectively [38]. Figure 6 shows the deconvoluted XPS spectra of C 1s and O 1s peaks of fresh GO. It confirms the presence of various oxygen functional groups such as carbonyl (C=O), carboxylic (−COOH), hydroxyl (C−OH) and ether (−C−O−C−), and $H_2O$ in GO surface [38]. We assume that the oxygen functional groups are also played a significant role in achieving the excellent surface morphology of Ni/GO nanocomposites. Figure 7d shows the Ni 2p XPS spectra of Ni/GO-1 and Ni/GO-2. New peaks in the Ni 2p region for Ni/GO-1 and Ni/GO-2 confirm the successful decoration of Ni with GO. The Ni $2p_{3/2}$ XPS peak at around 854.2 eV and the O 1s XPS peak at around 530 eV are from $Ni^{2+}$ and are associated with the Ni-O octahedral bonding of cubic rock salt NiO [39]. Moreover, satellite peaks were noticed at 861 eV (Ni $2p_{3/2}$ peak) which is due to shakeup process in the NiO structure [40]. It was noted that the XPS Ni $2p_{3/2}$ spectrum of Ni/GO at around 854 eV is closely similar to that reported previously by Grosvenor and co-workers [39]. The additional broad surface peak at around 858 eV reveals the possibility of mixed $Ni^{2+}/Ni^{3+}$ phase. Overall, the XPS result confirms that the Ni nanoparticles supported on GO surface are in the form of $Ni(OH)_2$, NiOOH, NiO, $Ni_2O_3$ and $NiO_2$. The broadening of O 1s peak of Ni/GO composites at around 531 eV is may be due the presence of C−O−Ni and C=O/O−Ni groups [41].

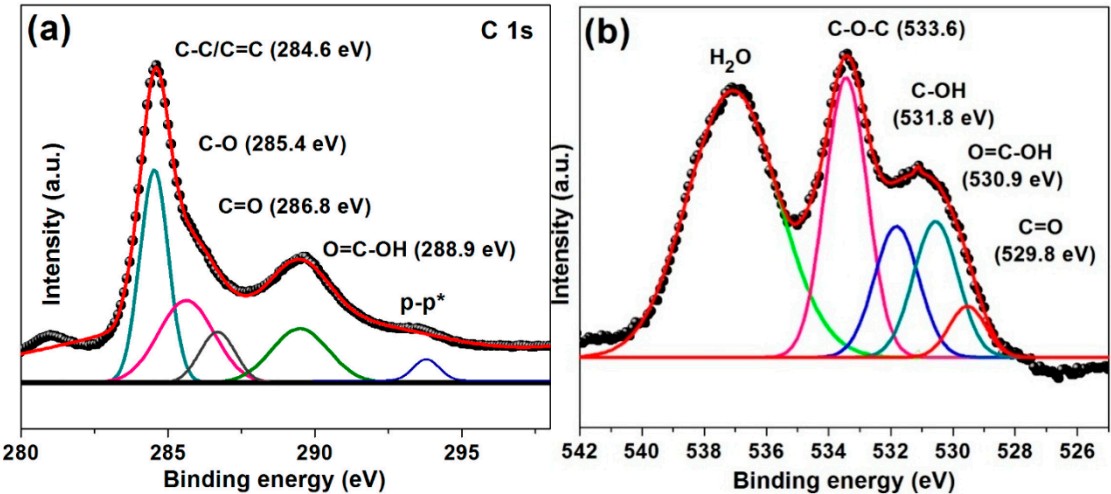

**Figure 6.** Deconvoluted XPS spectra of (**a**) C 1s and (**b**) O 1s peaks of fresh GO.

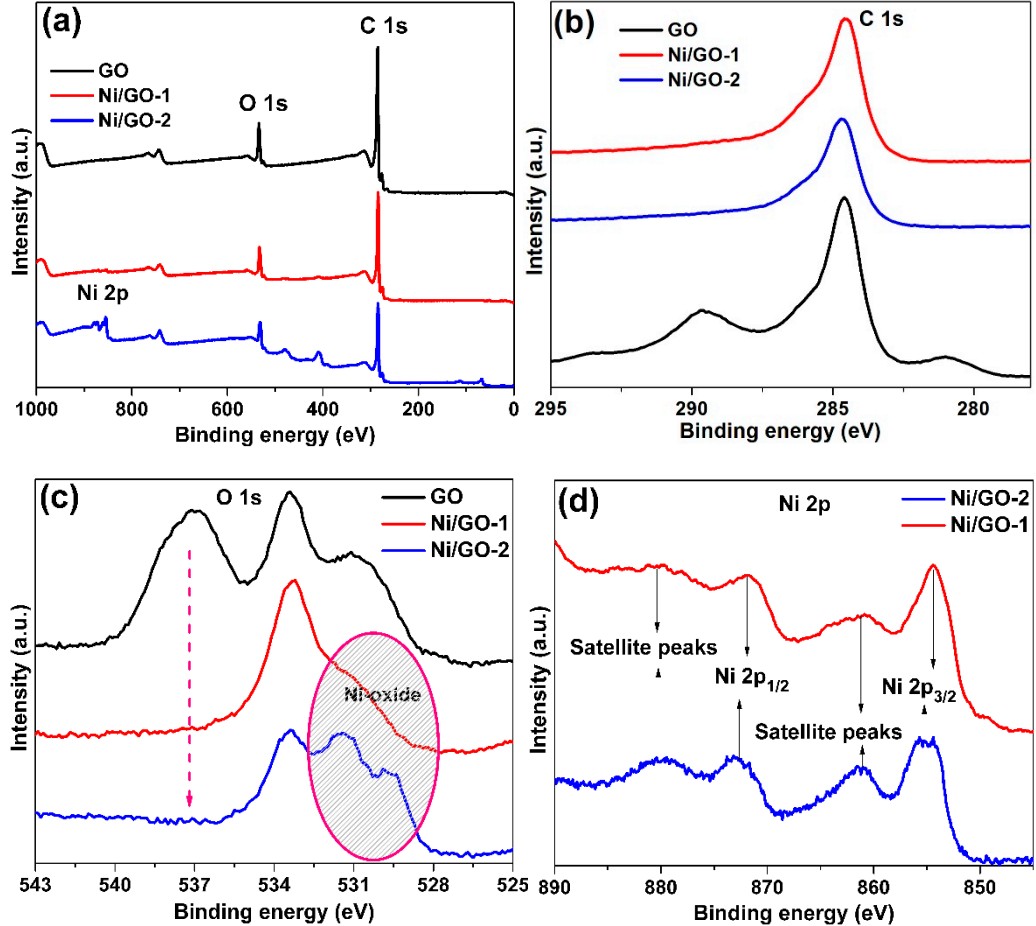

**Figure 7.** XPS spectra of GO, Ni/GO-1 and Ni/GO-2; (**a**) survey spectra, (**b**) C 1s peak, (**c**) O 1s peak and (**d**) Ni 2p peaks.

BET surface area, average pore volume and average pore size were determined for GO, Ni/GO-1 and Ni/GO-2 (Figure 8). Based on the results, the specific surface areas of GO, Ni/GO-1 and Ni/GO-2 were calculate to be 767.3, 333.1, and 301.4 m$_2$/g, respectively. A Type IV isotherm is noticed in Figure 8, confirming the presence of mesopores in the Ni/GO nanocomposites. Pore volumes of 1.289758, 0.457779 and 0.368079 cm$^3$/g were calculated for GO, Ni/GO-1 and Ni/GO-2, respectively. Similarly, pore size of GO, Ni/GO-1 and Ni/GO-2 was determined to be 7.18, 5.83 and 4.59 nm, respectively. Specific surface area of Ni/GO nanocomposites is very essential for an excellent energy and catalytic applications. Generally, GO nanocomposites often offer low BET surface area compare to raw GO [42]. Alike, the specific surface area of GO/Ni nanocomposites decreases with increasing amount of the Ni loading. In fact, the pores present in the GO could be occupied by the Ni nanoparticles [36]. However, the surface area of Ni/GO-1 and Ni/GO-2 was still shown to be larger compared to previously reported results; this shows the merits of the present preparation method. The better surface area of the Ni/GO nanocomposites may be due to the prevention of face-to-face aggregation of graphene layers by the Ni nanoparticles [36].

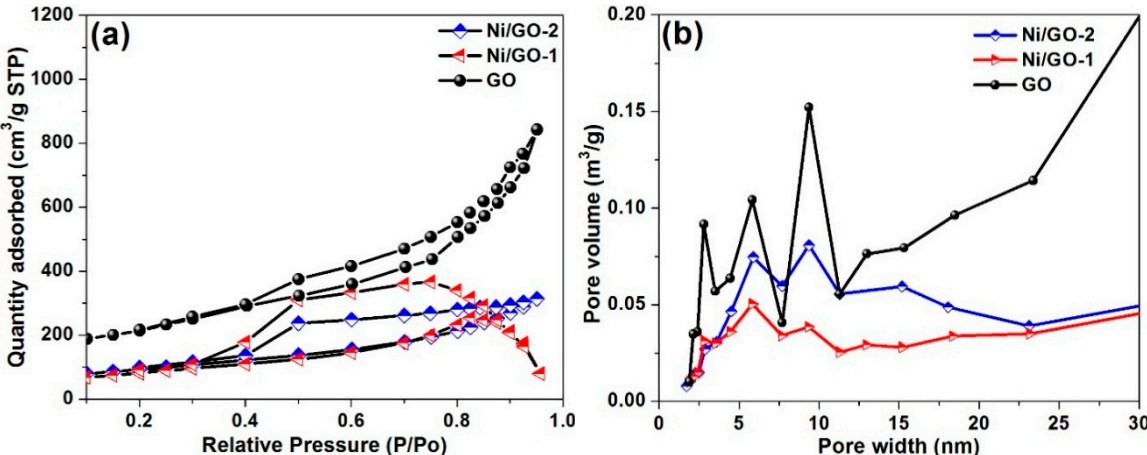

**Figure 8.** (**a**) Typical $N_2$ adsorption/desorption isotherm curves of BET isotherm and (**b**) pore volume of GO, Ni/GO-1 and Ni/GO-2.

### 3.2. Catalytic Conversion of 4- and 2-Nitrophenol

Catalytic conversion of nitrophenols (NP) to aminophenols (AP) is one of the most important reactions in green chemistry [43]. Nitrophenols and derivatives are common byproducts from the manufacture of synthetic dyes, herbicides and pesticides, and as a result, they are often found in environmental water and agricultural soil [44]. They are highly soluble and stable, and remain in water and soil for long time [45]. Moreover, long standing of nitrophenols in soil or water may cause serious harmful effects to animals, human beings and agricultural plants; therefore, the U.S. Environmental Protection Agency is listed the nitrophenols in the top 114 organic pollutants [46]. Whereas aminophenols are proved to be an important intermediate in preparing analgesic antipyretic drugs (such as phenacetin, paracetamol, and so on) and used as a suitable reducing agent for photographic developers and the dye industry [47]. Conversion of harmful such organic waste, nitrophenols, to valuable intermediate chemicals, aminophenols, in the presence of metal catalysts is very simple, greener and most efficient. Zhao et al. [48] prepared Pt–Ni alloy/reduced graphene oxide nanocomposite (Pt-Ni/r-GO) and used for the reduction of 4-nitrophenol to 4-aminophenol. They found that the Pt-Ni/r-GO is highly active which showed an excellent 94% conversion of 4-NP to 4-AP within 18 min. Bimetallic Ag-Au alloy nanoparticles supported on reduced graphene oxide nanocomposite (Au-Ag/r-GO) was obtained by Hareesh and coworkers [49]. The Au-Ag/r-GO demonstrated a complete conversion of 4-NP to 4-AP within 360 s and the reaction rate constant ($k_{app}$) was calculated to be $6.83 \times 10^{-3}$ $s^{-1}$. Similarly, GO supported nanocomposites such as AgNPs-rGO [50], AuNPs-RGO [51], PdNiP/RGO [52], NiNPs/Silica [53] and RGO-ZnWO$_4$-Fe$_3$O$_4$ [54] were reported for the reduction of nitrophenol. It should be noted that the most of the active nanocomposites are either bimetallic or noble metal-based catalysts. To our delight, the present mechanochemical preparation of mono metallic Ni/GO composites is a very simple, cost-effective and highly active for the conversion of NP to AP.

In the present study, catalytic property of Ni/GO nanocomposites was investigated. Aqueous phase reduction of nitrophenol to aminophenol by NaBH$_4$ was employed as a model reaction. Both 2-nitrophenol (2-NP) and 4-nitrophenol (4-NP) are taken for the investigation and UV-visible spectroscopy was used to evaluate the catalytic performance. Initially, the reaction conditions, such as catalyst, amount of catalyst, concentration of nitrophenol, concentration of NaBH$_4$, and reaction time, were optimized. We found no reaction in the absence of Ni/GO nanocomposites (Figure S3). However, upon the addition of aqueous NaBH$_4$ to 2-NP solution, the adsorption band was observed to be red-shifted (from 350 nm to 416 nm) which confirms the formation of 2-nitrophenolate ion. Similarly, the formation of 4-nitrophenolate ions by mixing of aqueous 4-NP and NaBH$_4$ was confirmed from the red-shift in the adsorption band of aqueous 4-NP at 317 nm (from 317 nm to 400 nm). Moreover, the adsorption maximum at 416 nm (2-nitrophenolate ion) and 400 nm (4-nitrophenolate ion) were

remained unaltered even after 10 h. To our delight, a very small the amount of catalyst is enough for the complete and rapid conversion of 2- and 4-NP. The required amount of Ni/GO-1 and Ni/GO-2 was found to be 0.75 and 2 mg respectively. However, it was found that the pure GO is inactive toward the reduction reaction. Subsequently, the concentration of nitrophenol and $NaBH_4$ was studied. As a result, 80 μL of 0.01 M 2-NP or 4-NP, and 4 mL of 0.015 M aqueous $NaBH_4$ were found to be the best condition to evaluate the catalytic performance of Ni/GO-1 and Ni/GO-2.

Figures 9 and 10 show the UV–vis spectra for the reduction of 2-NP and 4-NP in aqueous solution recorded every 30 s using different amount of Ni/GO-1 (1.0, 1.5 and 2.0 mg) and Ni/GO-2 (0.25, 0.50 and 0.75 mg). In addition, the plots of $\ln[C_t/C_0]$ versus reaction time for the reduction of 2-NP and 4-NP with $NaBH_4$ over Ni-GO-1 and Ni/GO-2 are also shown. The kinetic reaction rate constant ($k_{app}$) was calculated from the slope of the $\ln(C_t/C_0)$ versus time liner curve. Results showed that the Ni/GO nanocomposites are highly active in the reduction of 2- and 4-NP. The rate of the reduction process is rapidly increased with the amount of the Ni/GO nanocomposites and time. A 0.75 mg of Ni/GO-2 catalyst require just 150 s for the complete reduction of 4-NP and 90 s for 2-NP. Similarly, with 0.75 mg of Ni/GO-2, the 2-NP was completely converted to 2-AP within 90 s. While slightly higher amount of Ni/GO-1 was required for the complete reduction of 2- and 4-NP, which is due to the difference in the Ni-loading. A 2.0 mg of Ni/GO-1 was found to be enough for the 100 % conversion of both 2-NP and 4-NP. To the best of our knowledge, this is the best efficient Ni-based graphenenanocomposites for the reduction of 2- and 4-NP reported to date.

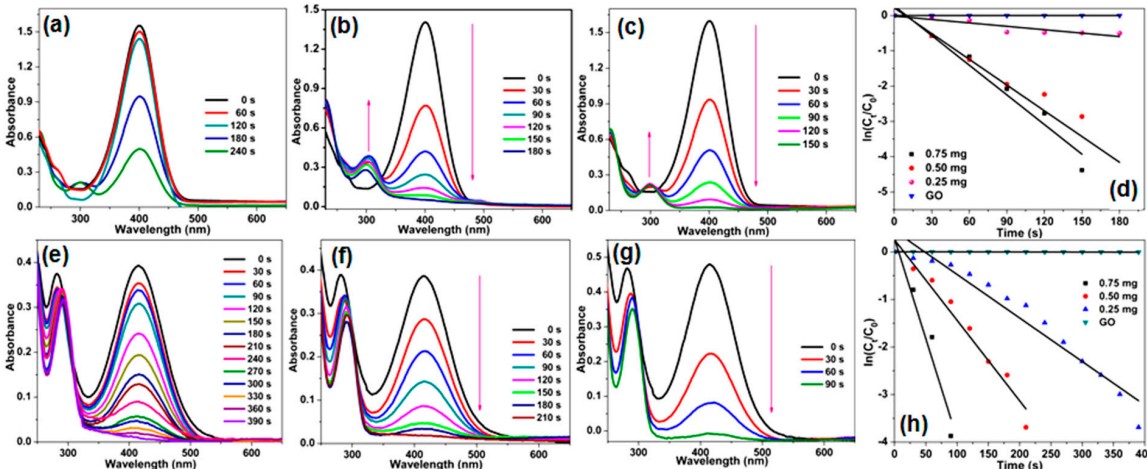

**Figure 9.** UV–vis spectra of the reduction of (**a–c**) 4-NP and (**e–g**) 2-NP in aqueous solution recorded every 30 s using different amount of Ni/GO-2 (0.25, 0.50 and 0.75 mg). Plots of $\ln[C_t/C_0]$ versus reaction time for the reduction of (**d**) 4-NP and (**h**) 2-NP with $NaBH_4$ over Ni/GO-2.

To further the reaction kinetics on the reduction of 2-NP and 4-NP by Ni/GO, nanocomposites were studied using the time-dependent UV-Vis spectra. The adsorption of nitrophenol molecules on the surface of GO can be ignored due to its inactiveness in reduction process. Similarly, the reaction rate is independent of $NaBH_4$ concentration due to its excess use. The linear correlation between $\ln(C_t/C_0)$ and time confirms the reduction reaction follows the pseudo-first-order reaction kinetics (Figure 9d,h and Figure 10d,h). From the slope of $\ln(C_t/C_0)$ *versus* time liner curve, the kinetic reaction rate constants ($k_{app}$) were calculated. The values were found to be excellent which show the catalyst is highly active and it reduces the 2- and 4-NP rapidly. For the Ni/GO-2 medicated reduction of 4-NP, the $k_{app}$ values were calculated to be $6.35 \times 10^{-3}$ $s^{-1}$ (0.25 mg), $30.9 \times 10^{-3}$ $s^{-1}$ (0.5 mg), and $35.4 \times 10^{-3}$ $s^{-1}$ (0.75 mg). Similarly, the $k_{appa}$ values for the reduction of 2-NP by Ni/GO-2 were $6.37 \times 10^{-3}$ $s^{-1}$ (0.25 mg), $12.8 \times 10^{-3}$ $s^{-1}$ (0.5 mg), and $70.7 \times 10^{-3}$ $s^{-1}$ (0.75 mg). The $k_{app}$ values were found to be slightly lower for the Ni/GO-1 which is due to the lower loading of Ni in Ni/GO-1 when compared to Ni/GO-1. The $k_{app}$ values of 5.09, 15.8 and $28.1 \times 10^{-3} s^{-1}$ were calculated for the GO/Ni-1 mediated

reduction of 4-NP with different catalyst amount of 0.3, 0.6 and 0.9 mg, respectively. Alike, for the GO/Ni-1 mediated 2-NP reduction process with different catalyst amount of 0.3, 0.6 and 0.9 mg, the $k_{app}$ values were calculated to be $9.23 \times 10^{-4}$ s$^{-1}$, 14.4 and $15.5 \times 10^{-3}$ s$^{-1}$. In addition, the reaction rate constant per unit mass ($k' = k_{app}/m$; where m-weight of the metal active site) was also calculated for a quantitative comparison. Surprisingly, an excellent $k'$ value of 47.2 and $94.3 \times 10^{-3}$ mg$^{-1}$ s$^{-1}$ was obtained for the reduction of 4-NP and 2-NP respectively, by Ni/GO-2 with 0.75 mg. Alike, the reduction of 4-NP and 2-NP respectively obtained $k'$ value of 14 and $7.5 \times 10^{-3}$ mg$^{-1}$ s$^{-1}$ by 2.0 mg of Ni/GO-1.

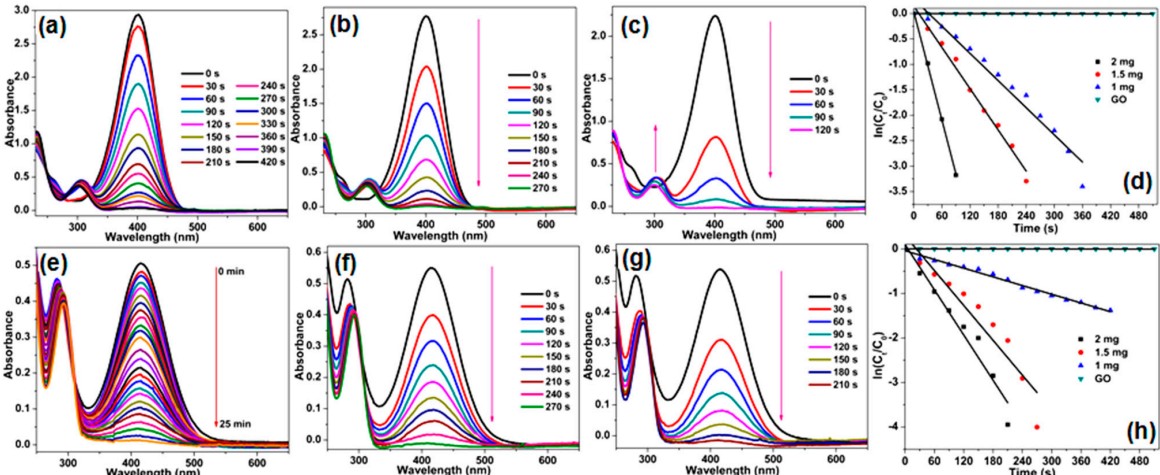

**Figure 10.** UV–vis spectra of the reduction of (**a**–**c**) 4-NP and (**e**–**g**) 2-NP in aqueous solution recorded every 30 s using different amount of Ni/GO-1 (1.0, 1.5 and 2 mg). Plots of ln[$C_t$/$C_0$] versus reaction time for the reduction of (**d**) 4-NP and (**h**) 2-NP with NaBH$_4$ over Ni/GO-1.

The present catalytic results were compared with previously reported results (Table 1). Silica nanotubes supported Ni nanocomposite (Ni/SNTs) with high Ni loading (23 wt%) demonstrated a complete reduction of 4-NP and the k$_{app}$ and k' values were calculated to be $84 \times 10^{-3}$ s$^{-1}$ and $91 \times 10^{-3}$ mg$^{-1}$ s$^{-1}$ respectively. Although the values are slightly higher than that of the present Ni/GO-1 and Ni/GO-2, the Ni/SNTs with 15.2 wt% of Ni loading gave lower k$_{app}$ and k' values were calculated to be $20 \times 10^{-3}$ s$^{-1}$ and $44 \times 10^{-3}$ mg$^{-1}$ s$^{-1}$, respectively, when compared to the present Ni/GO catalysts. It is worth noting that the Ni/GO-1 with just 3 wt% of Ni loading and Ni/GO-2 with just 8 wt% of Ni loading are better than the Ni/SNTs with 23 wt% of Ni content. Similarly, Ji and co-workers prepared reduced graphene oxide supported Ni catalyst (RGO/Ni) by wet synthesis method using hydrazine hydrate as reducing agent. The RGO/Ni gave k$_{app}$ and k' values of $0.25 \times 10^{-3}$ s$^{-1}$ and $0.04 \times 10^{-3}$ mg$^{-1}$ s$^{-1}$ respectively. In comparison to the above RGO/Ni catalyst, the present catalysts showed about 330-fold higher k$_{app}$ and k' values. We conclude that the present catalytic method is highly efficient for the preparation of Ni-graphene nanocomposites. In fact, the mechanical ball milling or etching are typical methods which can create more defect sites in graphene [28]. Moreover, the high surface area of catalysts, small size, homogenous dispersion of Ni nanoparticles and strong metal-support interaction, are the driving force of the present Ni/GO catalysts. Similarly, as listed in Table 1 the Ni/GO catalysts found to be better than the Pt–Ni/RGO [48], Au-Ag/r-GO [49], AgNPs-rGO [50], AuNPs-RGO [51], PdNiP/RGO [52], NiNPs/Silica [53], RGO-ZnWO$_4$-Fe$_3$O$_4$ [54], Ni/SNTs [55], Ni/MC-950 [56], and RGO/Ni [23].

**Table 1.** Comparison of present Ni/GO nanocomposites over other heterogeneous catalysts.

| S. No | Catalyst (amount used, mg) | Reactant | $k_{app}$ ($\times 10^{-3}$ s$^{-1}$) | $k'$ ($\times 10^{-3}$ mg$^{-1}$ s$^{-1}$) | TOF [a] (s$^{-1}$) | References |
|---|---|---|---|---|---|---|
| 1 | 23 wt.% Ni/SNTs | 4-NP | 84.0 | 91.0 | - | [55] |
| 2 | 15 wt.% Ni/SNTs | 4-NP | 20.0 | 44.0 | - | [55] |
| 3 | Ni/GO-1 (0.75) | 4-NP | 28.1 | 14.0 | 31.66 | This work |
| 4 | Ni/GO-2 (0.75) | 4-NP | 35.4 | 47.2 | 25.33 | This work |
| 5 | Ni/GO-1 (0.75) | 2-NP | 15.5 | 7.5 | 18.09 | This work |
| 6 | Ni/GO-2 (0.75) | 2-NP | 70.7 | 94.3 | 42.22 | This work |
| 7 | Ni/MC-750 (3) | 4-NP | 6.26 | 20.9 | 1.44 | [56] |
| 8 | RGO/Ni (6.5) | 4-NP | 0.25 | 0.04 | - | [57] |
| 9 | Pt–Ni/RGO (3) | 4-NP | 3.70 | 1.23 | 110.9 | [48] |
| 10 | Au-Ag/r-GO (0.1) | 4-NP | 3.47 | 34.7 | 0.042 | [49] |
| 11 | AgNPs-rGO | 2-NP | 0.44 | - | - | [50] |
| 12 | AuNPs-RGO (0.05) | 4-NP | 28.37 | 11.2 | 0.222 | [51] |
| 13 | PdNiP/RGO (3) | 4-NP | 23.51 | 7.7 | - | [52] |
| 14 | NiNPs/Silica (3) | 4-NP | 2.82 | 0.57 | - | [53] |
| 15 | RGO-ZnWO$_4$-Fe$_3$O$_4$ | 4-NP | 176.8 | 353.6 | - | [54] |
| 16 | Ni/SNTs (4) | 4-NP | 2.7 | 2.6 | - | [55] |
| 17 | Ni/MC-950 | 4-NP | 2.4 | 3.4 | 1.43 | [56] |
| 18 | RGO/Ni (6.5) | 4-NP | 14.8 | 2.3 | - | [23] |
| 19 | Ni/GNS | 2-NP | 3.06 | 1.53 | 0.31 | [41] |
| 20 | Ni/HHP | 2-NP | 69.1 | 27.6 | 0.03 | [57] |

[a] TOF, s$^{-1}$: (turnover frequency) moles of 4-NP converted per mole surface Ni per second.

The possible mechanisms and reusability of Ni/GO catalysts for the reduction of 2-NP and 4-NP were studied. In the reduction process, the BH$^{4-}$ from NaBH$_4$ and nitro group of 4-NP or 2-NP acted as donor and acceptor, respectively. At first, the nitrophenolate ion adsorbs on Ni surface of Ni/GO and produces active hydrogen atoms and subsequently reduces the nitrophenol to aminophenol [36,58]. The rapid reduction of nitrophenols is due to the fact that the electron sources in the catalytic system can be rapidly transferred from BH$_4^-$ to the Ni surface with the help of graphene, which provides an excellent interfacial electron relay medium. Figure 11 shows the reusability test of Ni/GO-1 and Ni/GO-2 for the reduction of both 2-NP and 4-NP. It was found that the Ni/GO catalyst can be reused several times without significant loss in its catalytic activity. Both Ni/GO-1 and Ni/GO-2 gave more than 95% of the catalytic products under the optimized reaction conditions. Overall, the Ni/GO nanocomposites are highly active for nitro-compound reduction, and their excellent reusability confirms theirenvironmental and economic benefits.

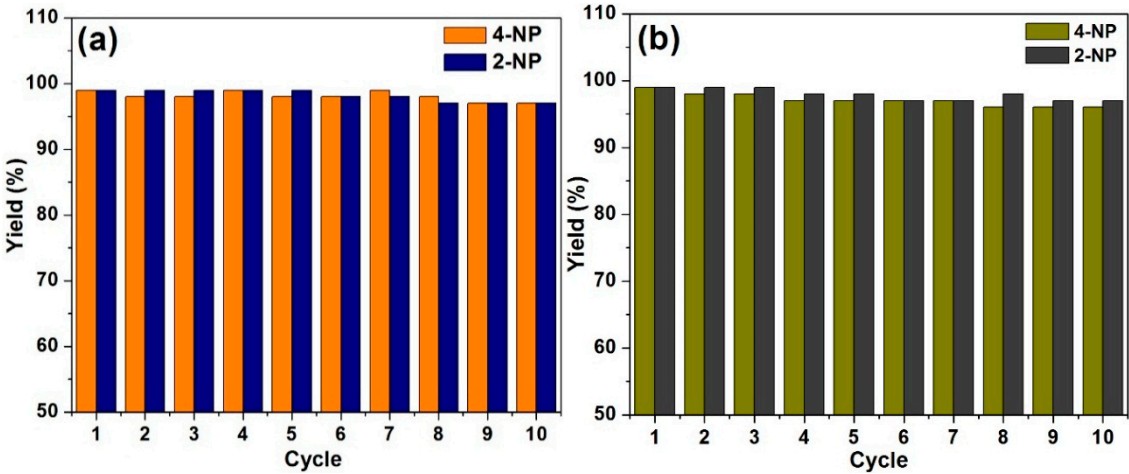

**Figure 11.** Reusability test of (**a**) Ni/GO-1 and (**b**) Ni/GO-2.

### 3.3. Electrochemical Studies

Inspired bythe excellent catalytic results, the present Ni/GO nanocomposites were applied for an energy stooge application. Indeed, the development of electrochemical energy storage systems (EES) is largely focused, due to its vital demand for clean and sustainable energy [59]. Most commercialized energy storage systems (EES) are batteries and supercapacitors [60]. In the present study, the capacitive performance of Ni/GO nanocomposites was investigated. Cyclic voltammetry (CV) curves were recorded for fresh GO, Ni/GO-1 and Ni/GO-2 in 6 M KOH electrolyte at different scan rates ranging from 5 to 50 mV in the potential range of −1.2 to 0.3 V. Results in Figure 12 show the CV curves of pure GO, Ni/GO-1 and Ni/GO-2. From the consequences, it can be clearly found that the Ni/GO nanocomposites exhibit higher capacitive current densities than pure GO. The CV curves of fresh GO is close to an ideal rectangle shape, which is caused by the electric double layer capacitor effect. The specific capacitance of fresh GO was calculated to be 67.87 F/g. To our delight, the NiO nanoparticles decorated GO nanocomposites showed excellent 5–7 fold enhancements in theirspecific capacitance. However, there wasno significant alteration in the CV curves of Ni/GO composites when compared to fresh GO. In general, the NiO-based electrode materials show redox peaks due to the redox reaction (NiO + OH⁻↔NiOOH + e⁻NiO + OH− ↔NiOOH + e⁻) [61]. However, the present case is different, and the close to an ideal rectangle shape of CV curves of Ni/GO composites indicates the predominant electric double layer capacitor effect. The specific capacitance of Ni/GO-1 and Ni/GO-2 was calculated to be 311.28 and 461.31 F/g, respectively. The enhanced capacitance of the Ni/GO nanocompositeswasmainly due to better/faster ion diffusion across the electrolyte-to-electrode interface and the surface layers of the electrochemically active Ni/GO materials. It shouldbe noted that the capacitance of Ni/GO nanocomposites is gradually increased with NiO loading. However, the Ni/GO with 15 wt% loading showed lower specific capacitance of 286 F/g compared to Ni/GO-2. Although the Ni loading is high, the high capacitance of Ni/GO-2 is due to its better and unique morphology.

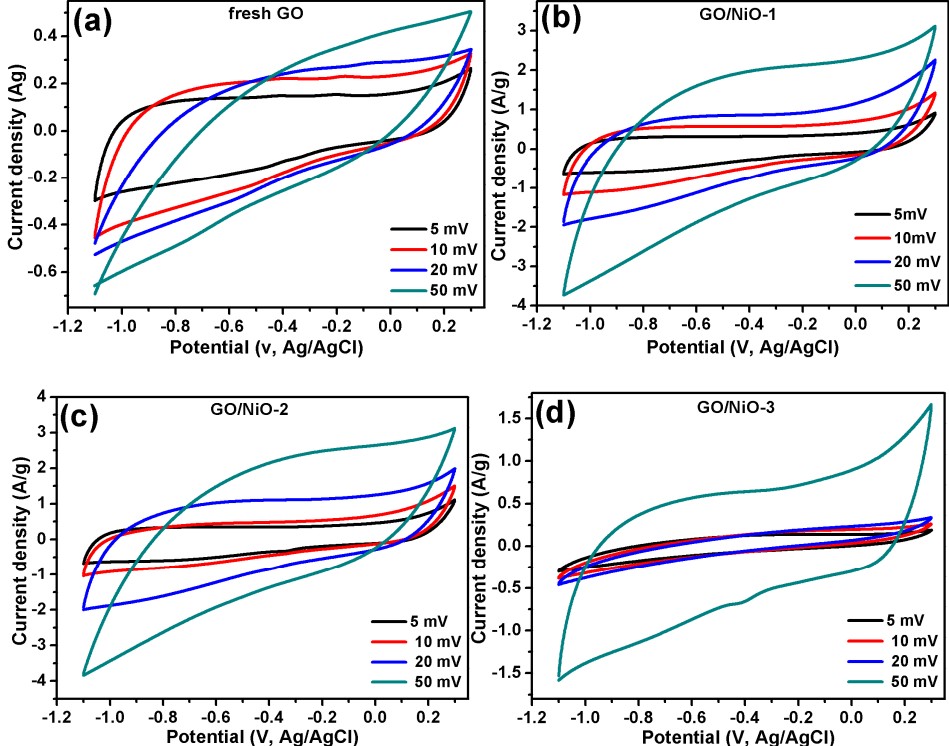

**Figure 12.** CV curves of (**a**) fresh GO, (**b**) Ni/GO-1, (**c**) Ni/GO-2 and (**d**) Ni/GO with 15 wt% of Ni at scan rates between 5 and 50 mVs⁻¹.

Electrochemical impedance spectroscopy (EIS) was conducted in order to further evaluate the electrochemical performance of GO, Ni/GO-1 and Ni/GO-2. The frequency range of 0.1 Hz to 100 kHz at open circuit potential with an AC perturbation of 5 mV was fixed. Figure 13b shows the Nyquist plots of GO, Ni/GO-1 and Ni/GO-2. It can be seen that the Nyquist plots of GO and Ni/GO composites are consist of a depressed arc in the high-frequency range (mainly contributes to the charge-transfer resistance of the redox reaction at the electrode/electrolyte interface) and a straight line in the low-frequency range (corresponds to the capacitor behavior in supercapacitance) [62]. The Nyquist plots of Ni/GO-1 and Ni/GO-2 show an oblique line instead of the theoretically-predicted vertical line relative to the axis, indicating the better ion diffusion across the electrolyte-to-electrode interface and the surface layers of the electrochemically active materials. This is due to the excellent properties of the Ni/GO nanocomposites such as high surface area, better dispersion of NiO nanoparticles and the unique morphology of the Ni/GO. In addition, the mechanochemical preparation of Ni/GO composites possess excellent contact between the graphene and the NiO nanoparticles nanoplates, which improves the charge-transfer kinetics and accelerate ion diffusion and faradic reaction, leading to enhanced rate performance and electrochemical activity.

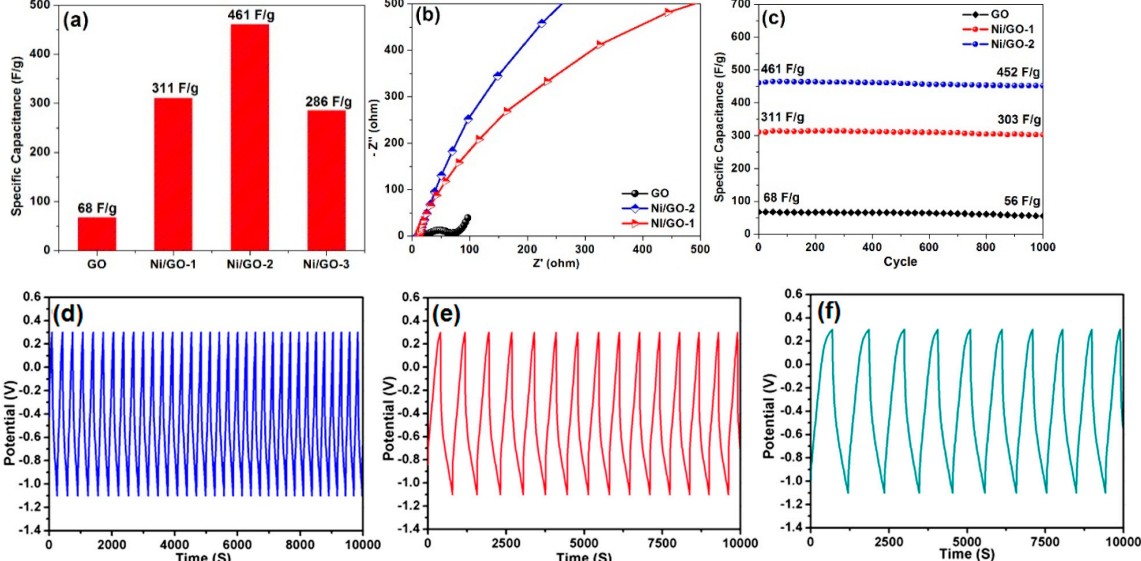

**Figure 13.** (**a**) Specific capacitance of GO, Ni/GO-1, Ni/GO-2 and Ni/GO-3, (**b**) Nyquist plots of GO, Ni/GO-1 and Ni/GO-2, (**c**) cycling stability of GO, Ni/GO-1 and Ni/GO-2, and galvanostatic charge–discharge behavior of (**d**) GO, (**e**) Ni/GO-1 and (**f**) Ni/GO-2.

The electrochemical performance of supercapacitors is highly dependent on cycling stability of the electrode material. The cycling performance of fresh GO, Ni/GO-1 and Ni/GO-2 were tested in 6 M KOH at scan rate of 5 mV. Figure 13c–e shows the recycling stability and the corresponding galvanostatic charge–discharge curves of GO, Ni/GO-1 and Ni/GO-2. Surprisingly, the electrode materials exhibited excellent cycling stability even after 1000 cycles. It is interesting to note that from the 1st cycle to 1000th cycle, the specific capacitance of the electrode material washighly maintained. Both Ni/GO-1 and Ni/GO-2 electrode materials maintained about 98% of capacitance retention, even after 1000 cycles, indicating excellent cycle stability. Overall, the results confirm that the mechanochemically-prepared Ni/GO nanocomposites are highly multifunctional, active and versatile.

## 4. Conclusions

In conclusion, an efficient and simple mechanochemical preparation method was developed for the preparation of Ni/GO nanocomposites. The morphology of Ni/GO nanocomposites can be tuned by simply varying the metal loading. The Ni/GO nanocomposites possess high specific surface

area and unique surface morphology. To our delight, the Ni-nanocomposites demonstrated superior catalytic performance toward the reduction of nitrophenols to aminophenols with high rate constant ($k_{app}$) of $35.4 \times 10^{-3}$ s$^{-1}$. In addition, the desirable environmental and economic benefits were realized thanks to their excellent reusability. To the best of our knowledge, this study presents one of the best Ni-based nanocomposites for the reduction of 2- and 4-NP reported to date. Surprisingly, the Ni/GO nanocomposites as an electrode material exhibited an excellent specific capacitance of 461 F/g in 6 M KOH at a scan rate of 5 mV. Electrochemical measurements confirmed the poor electrical resistance and high stability (no significant change in the specific capacitance even after 1000 cycles) of the Ni/GO nanocomposites. Overall, the results demonstrate that mechanochemically-prepared Ni/GO nanocomposites are highly active, multifunctional and versatile.

**Supplementary Materials:** The following are available online at http://www.mdpi.com/2073-4344/9/5/486/s1, Figure S1: TEM images and Ni/GO-3 with Ni wt% of 15, Figure S2. AFM images of Ni/GO with 15 wt% of Ni (Ni/GO-3); performed directly on the surface of samples and their three-dimensional projections, Figure S3. UV-vis spectra of (**a**) 2-NP and (**b**) 4-NP before and after adding NaBH4 solution.

**Author Contributions:** Conceptualization and methodology, M.G. and I.M.C.; Formal analysis, S.S.; Software and formal analysis, D.D. and M.G.; Data curation and investigation, M.G. and D.D.; Original draft writing and review and editing, M.G., I.M.C. and I.S.K.; Supervision, A.I., I.M.C. and I.S.K.

**Funding:** This research received no external funding.

**Acknowledgments:** This study was supported by Konkuk University KU research professor program.

**Conflicts of Interest:** Authors declare there is no conflict of interest.

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
