# Peer review of "Facile Mechanochemical Synthesis of Nickel/Graphene Oxide Nanocomposites with Unique and Tunable Morphology: Applications in Heterogeneous Catalysis and Supercapacitors"

_catalysts, doi:10.3390/catal9050486_

Round 1
Reviewer 1 Report
The authors synthesized Ni/GO catalysts from mechanochemical preparation method for the reduction of 2- and 4-nitrophenol and application in supercapacitor. They study the effect of metal loading (from 3w% to 15wt%) on particle size distribution, morphology and catalytic activity, using multiple characterization techniques. Results and discussion are well documented. However, there are some aspects which need to be revised.
1) Ni/GO-2 shows small particles of 3nm and large particles of 20 nm. But this catalyst exhibits the best catalytic activity and capacitive performance comparing with Ni/GO-1 and Ni/GO-3. It is not clear that the best performance is from 3 nm Ni or 20 nm Ni particles. The authors should clarify that.
2) It is well-known that smaller particles demonstrate higher specific activity because of higher exposed metal atoms. It turns out that the best performance of Ni/GO-2 may probably be from 3nm Ni particles. If it is ture, the appearance of chunky particles (26 nm) from the preparation method decrease utility of Ni.
3) In table 1, catalytic activity is not correctly compared between samples because TOFs are not reported.
4) Size distribution of Ni/GO-3 should be provided.
4) In Figure 5b, XRD patterns are similar to metallic Ni. The authors should provide the standard pattern of metallic Ni as well. If the peaks are metallic Ni, they should explain why.
5) In line 232 of pg 8, the authors should carefully deconvolute the XPS peaks. If there are C–O–Ni and C=O/O–Ni groups in O 1s, there should be corresponding peaks in C1s.
6) In line 247 of pg 10, the authors should provide in more detail why the introduction of Ni particles substantially decrease the surface area of GO.
7) In line 250 of pg 10, references should be added if the authors are comparing to other results.
8) The loading mass of the active material is not expounded, since it is a main factor affecting the capacitive performance of the electrode.
9) On page 4 line 143, “The galvanostatic charge–discharge measurement was performed at 0.5-10 A g-1 over a voltage 143 range of −0.4–0.6 V vs. Ag/AgCl”, but the data is missing.
10)In CV curve of a electrode with capacitive property, the current versus scan rate obeys the power law of i=avb. In Fig. 11, do the CV curves of the electrodes obey the law? Since the curves are similar for GO/NiO-3 electrode at 5, 10, 20 mV s-1, why much larger curve was obtained at 50 mV s-1?
11) How do the authors come to a conclusion of poor electrical resistance for the Ni/GO electrodes through the EIS measurement?
Author Response
The authors synthesized Ni/GO catalysts from mechanochemical preparation method for the reduction of 2- and 4-nitrophenol and application in supercapacitor. They study the effect of metal loading (from 3w% to 15wt%) on particle size distribution, morphology and catalytic activity, using multiple characterization techniques. Results and discussion are well documented. However, there are some aspects which need to be revised.
1) Ni/GO-2 shows small particles of 3nm and large particles of 20 nm. But this catalyst exhibits the best catalytic activity and capacitive performance comparing with Ni/GO-1 and Ni/GO-3. It is not clear that the best performance is from 3 nm Ni or 20 nm Ni particles. The authors should clarify that.
Ø Obviously the best catalytic and capacitive performance of Ni/GO-2 is mainly due to the small particle size of NiO. On the other hand, the relatively big size of 26 nm particles would have supported from the prevention of the face-to-face aggregation of graphene layers (this is also the main reason for the high specific surface area). It should be noted that the mean size of the Ni nanparticles is found to be lower when compared to the Ni nanparticles size of the Ni/GO-1 and Ni/GO-3. Some of the key points such as small particle size are highlighted in the revised manuscript.
2) It is well-known that smaller particles demonstrate higher specific activity because of higher exposed metal atoms. It turns out that the best performance of Ni/GO-2 may probably be from 3nm Ni particles. If it is ture, the appearance of chunky particles (26 nm) from the preparation method decrease utility of Ni.
Ø Yes we agree with the reviewer’s point, the best catalytic and capacitive performance of Ni/GO-2 is mainly due to the small particle size of NiO. The chunky particles of 26 nm particles would have helped to prevent the face-to-face aggregation of graphene layers (reason for the high specific surface area).
3) In table 1, catalytic activity is not correctly compared between samples because TOFs are not reported.
Ø As per the reviewer suggestion, the TOF values and the amount of catalyst used are now added in the revised manuscript. Please refer Table 1 in the revised manuscript.
4) Size distribution of Ni/GO-3 should be provided.
Ø Now it is provided in the supporting information. Please see Fig. S1 (d).
4) In Figure 5b, XRD patterns are similar to metallic Ni. The authors should provide the standard pattern of metallic Ni as well. If the peaks are metallic Ni, they should explain why.
Ø It has been included in the revised manuscript. The peak position and crystal planes of metallic Ni is given and compared with the Ni-oxide.
5) In line 232 of pg 8, the authors should carefully deconvolute the XPS peaks. If there are C–O–Ni and C=O/O–Ni groups in O 1s, there should be corresponding peaks in C1s.
Ø We agree with the reviewer’s point. In order to avoid confusion, we have now modified the XPS section in the revised manuscript.
6) In line 247 of pg 10, the authors should provide in more detail why the introduction of Ni particles substantially decrease the surface area of GO.
Ø The pores present in the carbon-support could be occupied by the metal nanoparticles. This is the main reason why the surface area of metal loaded-GO decreased when compared to fresh GO. These points have been added in the revised manuscript. Appropriate reference is also cited.
7) In line 250 of pg 10, references should be added if the authors are comparing to other results.
Ø As per the reviewer suggestion, appropriate reference has been cited.
8) The loading mass of the active material is not expounded, since it is a main factor affecting the capacitive performance of the electrode.
Ø It is provided in the experimental section. “. In a typical preparation of working electrode, a mixture of 1 mg of Ni/GO nanocomposites, 20 μL Nafion solution (5 wt%) and 400 μL isopropanol was sonicated for 2 h at room temperature. Then a 45 μL of the above mixture was taken and carefully deposited on the active area of the glassy carbon electrode. Finally, the glassy carbon electrode was kept in an oven at 80 °C for 30 min to remove the solvent.”
9) On page 4 line 143, “The galvanostatic charge–discharge measurement was performed at 0.5-10 A g-1 over a voltage 143 range of −0.4–0.6 V vs. Ag/AgCl”, but the data is missing.
Ø This mistake is now corrected in the revised manuscript.
10) In CV curve of a electrode with capacitive property, the current versus scan rate obeys the power law of i=avb. In Fig. 11, do the CV curves of the electrodes obey the law? Since the curves are similar for GO/NiO-3 electrode at 5, 10, 20 mV s-1, why much larger curve was obtained at 50 mV s-1?
Ø We noted it; we assumed that the morphology of the Ni/GO-3 may be the reason for the unusual behavior.
11) How do the authors come to a conclusion of poor electrical resistance for the Ni/GO electrodes through the EIS measurement?
Ø Though don’t have direct evidence for this claim, however, from the Nyquist plots of GO and Ni/GO composites support this claim.
Reviewer 2 Report
Authors reported nickel oxides/graphene oxide nano composites as catalysts for the reduction of 2- and 4-nitrophenols and as electrode materials for super capacitor applications. Their performances are very interesting, but this manuscript has to be revised with professional XPS analysis. Comments and questions are the following.
1. Abstract
Ni/GO-1 and Ni/GO-2 have to be changed to 3 wt.% NiO/GO and 8 wt.% NiO/GO, respectively, as their numbering doesn't provide any meaning. They already confirmed their Ni loading amounts and also mentioned the NiO nanoparticles are deposited on GO. All of them causes confusion.
2. Is there any reason for choosing 3 and 8 wt.%?
3. Any tick numbers are required for the y axis (Intensity, %) in Figure 2(g-i)
4. Any size distribution of 15 wt.% catalyst would be very helpful to explain the features in Figure 11(d) and other data in this manuscript.
5. In Figure 4, it is strongly recommended to separate SEM images and the map data because their over-layered image format is very common and better for the comparison.
6. Element analysis from EDS analysis is preferred to be listed in a table. What was the difference between average contents of 5 small areas vs. one big area.
7. Their XPS analysis is poor. They assigned important components of carbonyl, carboxylic, hydroxyl, ether, and water with neither deconvolution process nor mentioning their peak positions. It is in appropriate. In addition, simply comparing their peak intensities is not acceptable in the data analysis because each sample would have different sampling volume. Only relative ratios are acceptable in XPS element analysis unless their peaks are normalized. Deconvolution process and relative atomic compositions using atomic sensitivity factors must be done for XPS analysis in this manuscript.
8. Authors analyzed only one component (NiO) of Ni 2p at 854.2 eV, but the Ni 2p peak at 854.2 eV isn't symmetric, indicating more than one components as reported in Refs. [39, 40]. Ref. [39] assigned only one of two Ni 2p components, while Ref. [40] interpreted Ni has two oxidation components, +2 and +3. The characterization of Ni species is very important in this manuscript, as they're related to the catalytic active sites and involved in the mechanism.Their catalysts seem to have another component than NiO. However their XPS data analysis is neither completed nor accurate.
9. Neither Ref. [39] nor [40] reported any O 1s spectrum, but authors insist the broadening of O 1s peak in Figure 6(c) is due to the presence of C-O-N and C=O/O-Ni groups. It isn't wrong. Also the shape change in C and O 1s spectra was understood as the results of strong attachment of NiO, but there are many other reasons for the shape changes. The strong interaction could be related to a higher binding energy rather than peak shape, which usually indicate additional components or any change in relative ratios between components.
10. In Figures 7 and 8, the symbols need to be matched for the same, especially they're confusing in 7(a) and 8(d).
11. In Table 1, the comparison of rate constants would be meaningless when the Ni loading amount isn't considered. Further detailed conditions have to be included.
12. On page 13, any reference is required to mention the mechanism of the reduction of 1-NP and 4-NP.
Author Response
Authors reported nickel oxides/graphene oxide nano composites as catalysts for the reduction of 2- and 4-nitrophenols and as electrode materials for super capacitor applications. Their performances are very interesting, but this manuscript has to be revised with professional XPS analysis. Comments and questions are the following.
1. Abstract
Ni/GO-1 and Ni/GO-2 have to be changed to 3 wt.% NiO/GO and 8 wt.% NiO/GO, respectively, as their numbering doesn't provide any meaning. They already confirmed their Ni loading amounts and also mentioned the NiO nanoparticles are deposited on GO. All of them causes confusion.
Ø We agree with the reviewer’s suggestion, the abstract has been not modified in the revised manuscript.
2. Is there any reason for choosing 3 and 8 wt.%?
Ø Yes, from our previous studies and results, we assumed that the metal loadings of 3, 8 and 15 wt% would be suitable for this study.
3. Any tick numbers are required for the y axis (Intensity, %) in Figure 2(g-i)
Ø Now the y-axis has been given in the revised manuscript.
4. Any size distribution of 15 wt.% catalyst would be very helpful to explain the features in Figure 11(d) and other data in this manuscript.
Ø As per the reviewer suggestion, it is provided in the revised manuscript. Please refer supporting information Fig. S1(d).
5. In Figure 4, it is strongly recommended to separate SEM images and the map data because their over-layered image format is very common and better for the comparison.
Ø As per the reviewer suggestion, the map data has been separated out in the revised manuscript.
6. Element analysis from EDS analysis is preferred to be listed in a table. What was the difference between average contents of 5 small areas vs. one big area.
Ø We agree with the reviewer’s comments. Since the nanoparticles are uniformly decorated on GO surface, there is no significant difference in the wt% of Ni when the EDS analysis was carried out on five different areas. In addition, the ICP-MS analysis was also done to prove the factual loading of Ni. However, if the reviewer still thinks that the data should be tabulated, we are happy to do it.
7. Their XPS analysis is poor. They assigned important components of carbonyl, carboxylic, hydroxyl, ether, and water with neither deconvolution process nor mentioning their peak positions. It is in appropriate. In addition, simply comparing their peak intensities is not acceptable in the data analysis because each sample would have different sampling volume. Only relative ratios are acceptable in XPS element analysis unless their peaks are normalized. Deconvolution process and relative atomic compositions using atomic sensitivity factors must be done for XPS analysis in this manuscript.
Ø We agree with the reviewer’s suggestions. In order to avoid the confusion, we have modified the XPS discussion in the revised manuscript. In addition, deconvolution has been performed for the C 1s and O 1s XPS peaks of fresh GO and the data have been provided. Discussion on the peak intensity is now revised appropriately.
8. Authors analyzed only one component (NiO) of Ni 2p at 854.2 eV, but the Ni 2p peak at 854.2 eV isn't symmetric, indicating more than one components as reported in Refs. [39, 40]. Ref. [39] assigned only one of two Ni 2p components, while Ref. [40] interpreted Ni has two oxidation components, +2 and +3. The characterization of Ni species is very important in this manuscript, as they're related to the catalytic active sites and involved in the mechanism. Their catalysts seem to have another component than NiO. However their XPS data analysis is neither completed nor accurate.
Ø We agree with the reviewer’s suggestions. In order to avoid the confusion, we have modified the XPS discussion in the revised manuscript.
9. Neither Ref. [39] nor [40] reported any O 1s spectrum, but authors insist the broadening of O 1s peak in Figure 6(c) is due to the presence of C-O-N and C=O/O-Ni groups. It isn't wrong. Also the shape change in C and O 1s spectra was understood as the results of strong attachment of NiO, but there are many other reasons for the shape changes. The strong interaction could be related to a higher binding energy rather than peak shape, which usually indicate additional components or any change in relative ratios between components.
Ø We agree with the reviewer’s suggestions. Now we have modified the XPS discussion in the revised manuscript. Discussion on the peak shape and intensity has been revised appropriately.
10. In Figures 7 and 8, the symbols need to be matched for the same, especially they're confusing in 7(a) and 8(d).
Ø We checked it in the manuscript. However, we feel that the symbols are very clear and not confusing. We request reviewer to explain the question more clearly, so that we are very happy to modify it.
11. In Table 1, the comparison of rate constants would be meaningless when the Ni loading amount isn't considered. Further detailed conditions have to be included.
Ø As per the reviewer suggestion, the TOF values and the amount of catalyst used are provided in the revised manuscript. Please refer Table 1 in the revised manuscript.
12. On page 13, any reference is required to mention the mechanism of the reduction of 1-NP and 4-NP.
Ø As per the reviewer suggestion, appropriate reference has been cited.
Round 2
Reviewer 2 Report
Authors revised the manuscript according to the reviewers' comments and suggestions, but still some of them are confusing. This manuscript isn't ready for publication yet. Especially, from their XPS analysis, they confirmed the nickel composition on graphene oxides are not pure metal nanoparticles, but nickel oxides and hydroxides. Exactly it isn't Ni nanoparticles anymore. Authors need to find another term to describe them correctly in the whole content. Otherwise, it would be very confusing. The current title looks nickel metal nanoparticles are deposited on graphene oxides, but their samples are definitely different from metal nanoparticles. It is very important to be changed. If so, Ni/GO won't be suitable for this manuscript. Comments and questions are the following.
1. The symbols and colors need to be matched to the sample names to avoid any confusion, but authors haven't figured it yet.
Figure 2: Ni/GO-1 (red), Ni/GO-2 (blue, pink)
Figure 5(a): Ni/GO-1 (blue), Ni/GO-2 (red)
Figure 5(b): Ni/GO-1 (red), Ni/GO-2 (black)
Figure 7: Ni/GO-1 (red), Ni/GO-2 (blue)
Figure 8(a): Ni/GO-1 (red/white, triangle), Ni/GO-2 (pink/white, diamond)
Figure 8(b): Ni/GO-1 (red, circle), Ni/gO-2 (black, circle)
Figure 13(b): Ni/GO-1 (blue, circle), Ni/GO-2 (red, circle)
Figure 13(c): Ni/GO-1 (red, circle), Ni/GO-2 (blue, circle)
Figure 13(d): (blue) for GO?
Figure 13(e): (red) for Ni/GO-1:
Figure 13(f): (green) for Ni/GO-2?
2. Table 1
TOF needs to be followed by the unit like TOF (S-1) on the top row, and only numbers are listed in other rows below. Besides, only 1 reference is listed for 2-NP. More is recommended.
3. Figure 6
XPS analysis isn't good yet. Shirley background is required for the deconvolution of their spectra, not linear. In addition, the relative area percentage and binding energy of each component need to be displayed in the figure like in Figure 5 of Ref. [38]. Those are very common in XPS data processing. By the way, the deconvolution results don't look to be matched to C 1s spectrum of 6(a) .It might be the reason why they put the spectrum above the components' peaks, not overlapped. It should be like 6(b).
4. Figure 7(b)
There are 3 red arrows, but nothing is mentioned.
5. Figure 7(c)
Authors assigned the peaks at around 531 eV for Ni oxides, but no reference is provided for O 1s spectra of nickel oxides. They already mentioned other species (COOH, CO, C=O) in Figure 6(b). If so, how could they be distinguished from them at the same peak positions. It must be explained reasonably.
6. Figure 13 (d,e,f)
Those would be the cycling stability results, but the unit is time (s), not cycle. It isn't clear why they have different time intervals. Are those for each electrode material, respectively, or only one?
7. Figure S3(a)
Two plots couldn't be recognized by the same black color. It should be changed such as S3(b)
Author Response
Reponses to Reviewers’ Comments
Comments and Suggestions for Authors
Authors revised the manuscript according to the reviewers' comments and suggestions, but still some of them are confusing. This manuscript isn't ready for publication yet. Especially, from their XPS analysis, they confirmed the nickel composition on graphene oxides are not pure metal nanoparticles, but nickel oxides and hydroxides. Exactly it isn't Ni nanoparticles anymore. Authors need to find another term to describe them correctly in the whole content. Otherwise, it would be very confusing. The current title looks nickel metal nanoparticles are deposited on graphene oxides, but their samples are definitely different from metal nanoparticles. It is very important to be changed. If so, Ni/GO won't be suitable for this manuscript. Comments and questions are the following.
Ø We agree with the reviewer’s suggestion. In the present study, the Ni supported on GO is a composition of Ni(OH)2, NiOOH, NiO, Ni2O3 and NiO2. It is little difficult describe it in a single code. In order to avoid the confusion, we have clearly mentioned it in the abstract and introduction section that “the nanocomposites (Ni/GO, where Ni is a composition of Ni(OH)2, NiOOH, NiO, Ni2O3 and NiO2), 3 wt.% NiO/GO (Ni/GO-1) and 8 wt.% NiO/GO(Ni/GO-2).” We hope this would be helpful to the readers to understand the work clearly without any confusion.
1. The symbols and colors need to be matched to the sample names to avoid any confusion, but authors haven't figured it yet.
Figure 2: Ni/GO-1 (red), Ni/GO-2 (blue, pink)
Figure 5(a): Ni/GO-1 (blue), Ni/GO-2 (red)
Figure 5(b): Ni/GO-1 (red), Ni/GO-2 (black)
Figure 7: Ni/GO-1 (red), Ni/GO-2 (blue)
Figure 8(a): Ni/GO-1 (red/white, triangle), Ni/GO-2 (pink/white, diamond)
Figure 8(b): Ni/GO-1 (red, circle), Ni/gO-2 (black, circle)
Figure 13(b): Ni/GO-1 (blue, circle), Ni/GO-2 (red, circle)
Figure 13(c): Ni/GO-1 (red, circle), Ni/GO-2 (blue, circle)
Figure 13(d): (blue) for GO?
Figure 13(e): (red) for Ni/GO-1:
Figure 13(f): (green) for Ni/GO-2?
Ø As suggested by the reviewer, the symbols and colors are now matched to the sample names to avoid the confusion. In the revised manuscript, we followed GO (black, circle), Ni/GO-1 (red, triangle), and Ni/GO-2 (blue, diamond).
2. Table 1
TOF needs to be followed by the unit like TOF (S-1) on the top row, and only numbers are listed in other rows below. Besides, only 1 reference is listed for 2-NP. More is recommended.
Ø The unit has been move to the top row in Table 1, and more references are listed for 2-NP in Table 1.
3. Figure 6
XPS analysis isn't good yet. Shirley background is required for the deconvolution of their spectra, not linear. In addition, the relative area percentage and binding energy of each component need to be displayed in the figure like in Figure 5 of Ref. [38]. Those are very common in XPS data processing. By the way, the deconvolution results don't look to be matched to C 1s spectrum of 6(a) .It might be the reason why they put the spectrum above the components' peaks, not overlapped. It should be like 6(b).
Ø As suggested by the reviewer, the fitting has been redone and the binding energy of each component has been noted in the Fig 6a.
4. Figure 7(b) There are 3 red arrows, but nothing is mentioned.
Ø It has been now corrected in the revised manuscript.
5. Figure 7(c)
Authors assigned the peaks at around 531 eV for Ni oxides, but no reference is provided for O 1s spectra of nickel oxides. They already mentioned other species (COOH, CO, C=O) in Figure 6(b). If so, how could they be distinguished from them at the same peak positions. It must be explained reasonably.
Ø An appropriate reference has been cited for the claim.
6. Figure 13 (d, e, f)
Those would be the cycling stability results, but the unit is time (s), not cycle. It isn't clear why they have different time intervals. Are those for each electrode material, respectively, or only one?
Ø It has been now corrected in the revised manuscript.
7. Figure S3(a)
Two plots couldn't be recognized by the same black color. It should be changed such as S3(b)
Ø It has been now corrected in the revised manuscript.